# Fossil evidence unveils an early Cambrian origin for Bryozoa

Zhiliang Zhang[1,2 ✉], Zhifei Zhang[1 ✉], Junye Ma[3], Paul D. Taylor[4], Luke C. Strotz[1 ✉], Sarah M. Jacquet[5], Christian B. Skovsted[6], Feiyang Chen[1,2,7], Jian Han[1] & Glenn A. Brock[1,2]

Bryozoans (also known as ectoprocts or moss animals) are aquatic, dominantly sessile, filter-feeding lophophorates that construct an organic or calcareous modular colonial (clonal) exoskeleton[1–3]. The presence of six major orders of bryozoans with advanced polymorphisms in lower Ordovician rocks strongly suggests a Cambrian origin for the largest and most diverse lophophorate phylum[2,4–8]. However, a lack of convincing bryozoan fossils from the Cambrian period has hampered resolution of the true origins and character assembly of the earliest members of the group. Here we interpret the millimetric, erect, bilaminate, secondarily phosphatized fossil *Protomelission gatehousei*[9] from the early Cambrian of Australia and South China as a potential stem-group bryozoan. The monomorphic zooid capsules, modular construction, organic composition and simple linear budding growth geometry represent a mixture of organic Gymnolaemata and biomineralized Stenolaemata character traits, with phylogenetic analyses identifying *P. gatehousei* as a stem-group bryozoan. This aligns the origin of phylum Bryozoa with all other skeletonized phyla in Cambrian Age 3, pushing back its first occurrence by approximately 35 million years. It also reconciles the fossil record with molecular clock estimations of an early Cambrian origination and subsequent Ordovician radiation of Bryozoa following the acquisition of a carbonate skeleton[10–13].

The Cambrian fossil record chronicles in exceptional detail the emergence of major bilaterian clades and continues to provide chronological constraints on the evolutionary diversification of disparate metazoans from a common ancestor[12–15]. Nearly all animal phyla, including soft-bodied Deuterostoma[14], Entoprocta[16], Phoronida[17] and Priapulida[12], made their first appearance during the Cambrian evolutionary radiation[12,13,18]. A key exception is the 'missing' colonial lophotrochozoan phylum Bryozoa, in which six of the eight recognized orders belonging to the classes Stenolaemata and Gymnolaemata appear abruptly with considerable diversity during the early Ordovician period[6,7,19,20]. Furthermore, there is a major time gap (approximately 44 million years) between the first fossil record of unequivocal bryozoans in the earliest Ordovician (Tremadocian)[4,7] and the deeper origination in the early Cambrian (Terreneuvian) estimated using modern molecular clock analyses[10–12,21].

Bryozoa is the most speciose of the lophophorate phyla firmly nested within Lophotrochozoa, characterized by iterated units (zooids) demonstrating hierarchical levels of modularity, and (apart from one genus) is the only exclusively colonial group of metazoans[1,22–24]. The key innovation of modularity initiated a novel pattern of colonial growth that led directly to a burst of morphological diversification and subsequent ecosystem proliferation, especially during the Great Ordovician Biodiversification Event[1,18,25,26]. Increased fossil sampling has gradually pushed back the oldest occurrence of bryozoans[19,20], most recently into the early Tremadocian[4], while the bryozoan affinity of the late Cambrian (Furongian) genus *Pywackia* remains highly debated[2,4,7,18]. Hence, a Cambrian origin for Bryozoa is not completely unpredicted and many authors have suggested a non-mineralized organic colony might explain the lack of a Cambrian record for the group[3–7,19,20].

Here we describe rare but exquisitely preserved specimens of a millimetric modular fossil, *Protomelission gatehousei*[9] from the early Cambrian of Australia and South China (Extended Data Fig. 1). Scanning electron microscopy (Figs. 1, 2, Extended Data Figs. 2, 3) and X-ray tomographic microscopy (Fig. 3, Extended Data Fig. 4) images reveal a combination of character traits that suggest a stem-group bryozoan affinity for *P. gatehousei* but distinguish the taxon from all extant and extinct clades. The interpretation of this secondarily phosphatized fossil from lower Cambrian rocks of South China and South Australia as a putative bryozoan indicates that modular bryozoans evolved synchronously with most other stem-group metazoans during the Cambrian evolutionary radiation[12].

## Results

The finely phosphatized millimetric colony of *P. gatehousei* is bifoliate, compressed, lacks bifurcation, tapers apically and has an

[1]State Key Laboratory of Continental Dynamics, Shaanxi Key Laboratory of Early Life & Environments and Department of Geology, Northwest University, Xi'an, China. [2]Department of Biological Sciences, Macquarie University, Sydney, New South Wales, Australia. [3]State Key Laboratory of Palaeobiology and Stratigraphy, Nanjing Institute of Geology and Palaeontology, CAS, Nanjing, China. [4]Department of Earth Sciences, Natural History Museum, London, UK. [5]Department of Geological Sciences, University of Missouri, Columbia, MO, USA. [6]Department of Palaeobiology, Swedish Museum of Natural History, Stockholm, Sweden. [7]School of Resources and Geosciences, China University of Mining and Technology, Xuzhou, China. ✉e-mail: zhiliang.zhang@mq.edu.au; elizf@nwu.edu.cn; lukestrotz@nwu.edu.cn

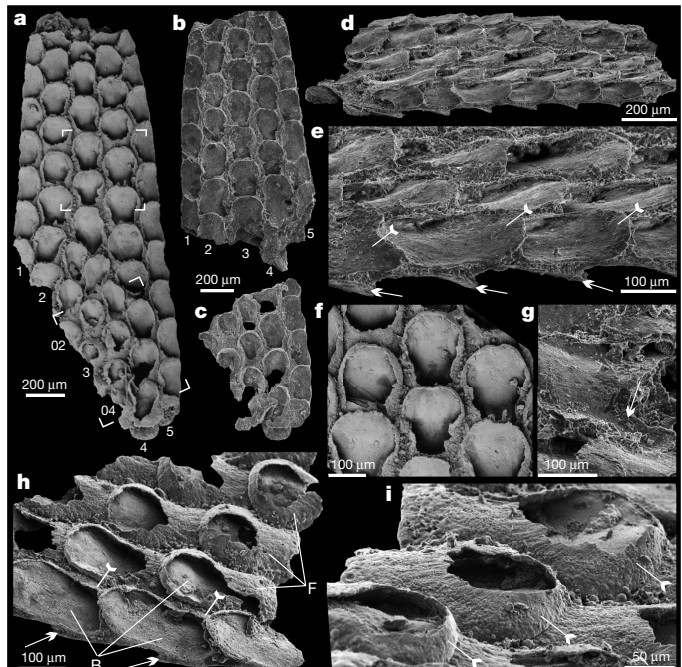

**Fig. 1 | *Protomelission gatehousei* from the Cambrian Wirrealpa Limestone, South Australia. a–g**, Holotype, SADME 10470. **a**, Front side of the colony originally published in ref. [9], noting the seven series of zooids. Top box corners indicate the area shown in **f**; bottom box corners show the broken-off part in **c**. **b**, The top broken part of **a**. **c**, The lower broken part of **a**. **d**, Oblique lateral view of the bilaminate colony. **e**, Enlarged view of **d**, showing the staggered budding pattern and the curved basal walls of the two back-to-back layers (arrows and tailed arrows) in the bifoliate colony. **f**, Quincuncial arrangement of sub-hexagonal zooids with broken frontal walls, originally published in ref. [9]. **g**, Lateral view of uncovered zooids; note the minute spoon-shaped structure (arrow) at the proximal end of basal wall extending backwards underneath the distal part of the parent zooid. **h**, **i**, SADME 10470-2. **h**, Lateral view of a broken colony, showing the largely broken frontal walls (tailed arrows) and basal walls of opposite layer (arrows). **i**, Enlarged view of three adjacent zooids. Note the dome shape of the distal part of frontal wall (tailed arrows), and almost circular orifice of zooid. B, basal wall; F, frontal wall.

elliptical holdfast at the base (Figs. 1a–c, 2a–c, 3a), suggesting an erect, self-supported colony anchored to the substrate (Extended Data Fig. 2a, b). Colonies range from 1.8–2.2 mm in height, 0.1–0.2 mm in thickness and 1.0–1.5 mm in width, which is very similar to the width of Ordovician erect bifoliate cryptostomes[27] (2.1 mm). The erect colony is composed of two layers of zooids (Figs. 1d–e, 2b, Extended Data Figs. 2c, 3c) arranged in simple linear series as back-to-back laminae[23,27] (Fig. 3a, d–g, Extended Data Fig. 4, Supplementary Videos 1-6).

**Fig. 2 | *Protomelission gatehousei* from the Cambrian Xihaoping Member, Dengying Formation, South China, ELI XYB 4 AN02. a**, Front side of the colony, noting the five series of zooids, box corners indicate the area shown in **d**. **b**, Oblique lateral view of the bifoliate colony, showing zooids in the back-to-back layers and the median mesotheca (arrow). **c**, Oblique basal view showing holdfast base and zooids of the opposite layer, box corners indicate the area shown in **f**. **d**, Quincuncial arrangement of hexagonal zooids; note spaces between adjacent zooids (arrows), frontal walls (tailed arrows) and basal walls. **e**, Lateral view showing the staggered pattern of zooids (arrows) in both layers, and a frontal wall on the margin (tailed arrow). **f**, Hexagonal zooids, showing the bases of the frontal walls (arrows). **g**, Enlargement of the fine wrinkles on the frontal walls, and granular phosphatized basal wall (arrow). B, basal wall.

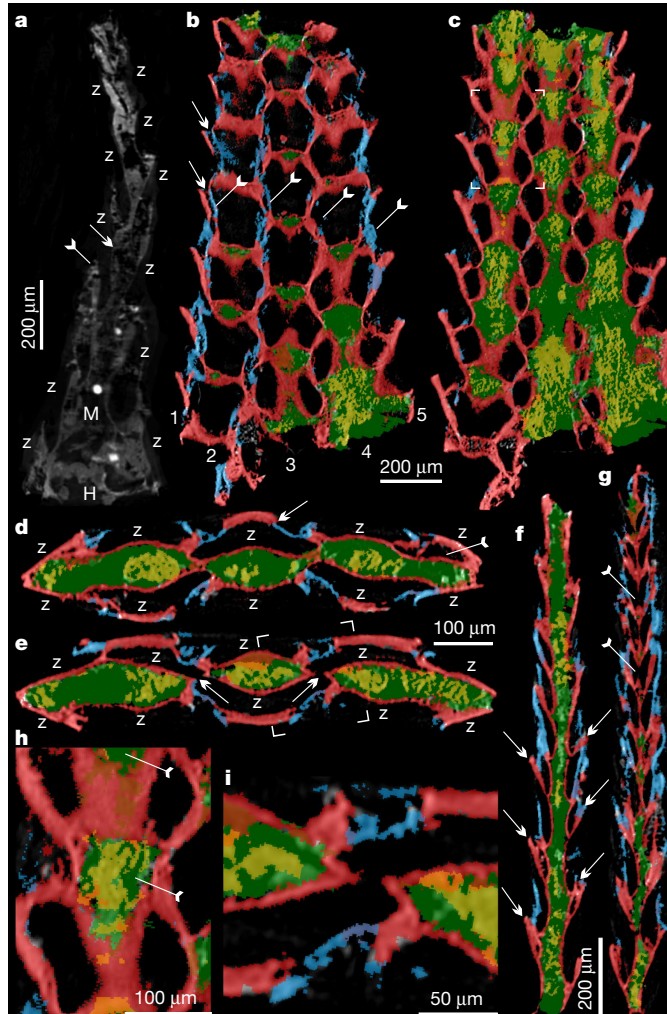

**Fig. 3 | X-ray tomographic microscopy images of *P. gatehousei*. a**, ELI XYB 4 AN02, longitudinal section. Frontal wall (tailed arrow) and basal wall (arrow) are indicated. **b–g**, SADME 10470. **b, c**, Tangential section. **b**, Five series of zooids and related four lines of frontal walls (tailed arrows). The space between adjacent zooids marked by arrows. **c**, Mesotheca/median lamina connected with above basal walls, box corners indicate the area shown in **h. d, e**, Transverse section. **d**, Zooids on both layers along with median mesotheca, noting new budding zooid (tailed arrow) and daughter basal wall overlapping parent frontal wall (arrow). **e**, Possible zooidal connection through the space of the median mesotheca (arrow), box corners indicate area shown in **i. f, g**, Longitudinal section, showing bilaminate pattern of zooids on the two back-to-back layers. **f**, Staggered pattern of zooids in both layers. The curved basal wall is indicated with arrows. **g**, Probable connections between adjacent zooids from back-to-back layers through space of median mesotheca, indicated by tailed arrows. **h**, Two pairs of zooids, magnified, showing spoon-shaped structures of parent and daughter zooids indicated by tailed arrows. **i**, Close-up of zooidal connection. Blue, frontal wall; green, mesotheca with secondary phosphatic cement in yellow; red, basal wall. H, holdfast; M, mesotheca; Z, zooid.

The zooids are sub-hexagonal in outline and flat box-shaped (Figs. 1f, h, 2d, Extended Data Fig. 3a–c). They are uniform in size, with an average width of 174 μm and length of 220 μm (Extended Data Table 1). There are up to 100 zooids in total in the bilaminate colony (Fig. 1a). Polymorphic differentiation of the zooids is absent, and there are no diaphragm-like structures in the zooids[7] (Figs. 1–3, Extended Data Figs. 2–4). Zooids are inclined at about 25° to the median lamina (mesotheca) (Fig. 3f, Extended Data Fig. 4a) and form a quincuncial pattern on the surface of the colonies (Figs. 1f, 2d, f, 3b, Extended

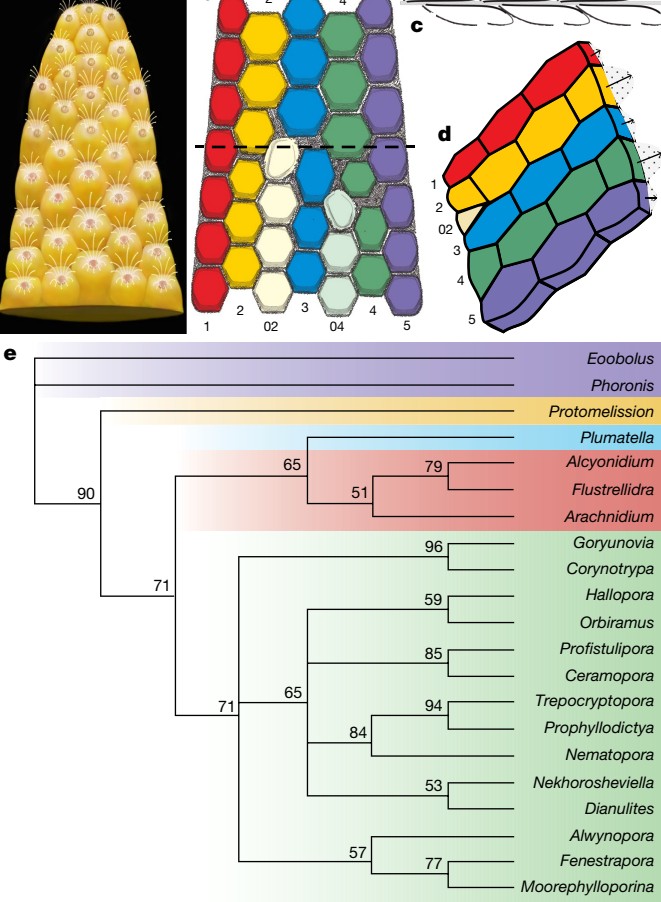

**Fig. 4 | Reconstruction and growth pattern of *P. gatehousei*, and its inferred phylogenetic relationships. a**, Front surface view, artwork created by X. Liu. **b**, Seven series of zooids, two of which terminate distally, resulting in five series; dashed line indicates the plane of sectioning in figure d. **c**, Budding process of two layers along the median mesotheca in longitudinal section. **d**, Distal zooidal bud formation[29] in six adjacent linear series, with termination of series-02. **e**, Fifty-percent-majority rule consensus phylogenetic tree inferred using morphological characters and Bayesian analysis based on a matrix of 21 taxa and 52 characters (see Methods and Supplementary Data 3, 4 for source data and additional information). Node values are Bayesian posterior probability support values. Coloured areas indicate the three taxonomic classes that comprise the Bryozoa, along with *P. gatehousei* and outgroups. Purple, outgroups; yellow, *Protomelission*; blue, Phylactolaemata; red, Gymnolaemata; green, Stenolaemata.

Data Fig. 3f), with 8–11 zooids within 2 mm longitudinally, and 7, 5 or 3 primary-order modular units of zooids arranged symmetrically on either side of the main median longitudinal axis (series-3) of the colony (Figs. 1a, 2a, 3b). The inflated frontal wall is thin and convex, imperforate, apically forming part of a hemispherical dome with a circular to ovoid opening in the best-preserved specimens (Fig. 1h, i, Extended Data Fig. 3a, b, f). Fine wrinkle structures developed on the frontal wall (Figs. 2g, Extended Data Fig. 2d, h) suggest an originally organic composition with labile and ductile properties, probably secreted by an underlying epithelium[2] and replicated during early diagenetic phosphatization (Extended Data Figs. 2i, j, 3h, j, k, 5c–k). The ultrastructure of the basal and frontal walls consists of diagenetic irregular apatite (Fig. 2g, Extended Data Figs. 2i, 3k), reflecting the secondary phosphatization of the original organic zooid body wall (Extended Data Fig. 5c–k). Spherulitic microstructures (Extended Data Figs. 2j, 3j) are also present and may be associated with microbially mediated phosphate replacement or diagenetic processes[28].

The developmental sequence of zooidal budding consists of five or seven alternating, back-to-back linear (longitudinal) series resulting in a palmate multiserial bilaminate colony (Figs. 3a–g, 4a). New zooids were budded at the distal tip of the colony, from pre-existing parents in an upward tapering growth vector (Figs. 1e, 3f, g, 4b, c). During clonal growth, the newly formed basal wall sequentially grew into contact with the walls of three adjacent zooids, entirely partitioning the original body (Fig. 3b, d, f, Extended Data Fig. 4a, f), demonstrating a zooidal budding process[1,29] (Fig. 4c, d, Extended Data Fig. 4a). As the colony grew apically, longitudinal module series of zooids on either side (series-02 and series-04) of the main median series-3 axis stopped budding to provide accommodation space for adjacent linear series of zooids to grow (Figs. 1a, 4b, d). As a consequence, the whole colony achieved a distally tapering morphology (Figs. 1a, 2a, 4a, b, Extended Data Fig. 5a, b). The exhalant currents of filtered water would probably have been vented out from the sharp colony edges by analogy with living bryozoans with palmate branches[1]. Compared with central series zooids, the marginal series (series-1 and series-5) demonstrate a relatively slow growth of zooids, which probably resulted from high-level control on the relative growth rates across different parts of the colony[1,29] (Figs. 1b, 4a, Extended Data Fig. 6a).

## Discussion

*Protomelission gatehousei* meets almost all recognition criteria expected in fossil bryozoans[2] (Extended Data Fig. 6b, Extended Data Table 2). The general morphology, zooid arrangement, budding direction and pattern are comparable to members of the Stenolaemata, which have been suggested to have been derived from a soft-bodied ctenostome-grade ancestor during the Cambrian[2,5,19,30]. With an originally unmineralized body-plan, phosphatized preservation and box-shaped zooids, and in keeping with its basal phylogenetic position (Fig. 4e), *P. gatehousei* shares traits with taxa from a number of classes within Bryozoa, including the soft-bodied Gymnolaemata (Ctenostomata)[19,20,30]. On the basis of phylogenetic analyses, we conclude that *P. gatehousei* potentially represents a stem-group bryozoan (Fig. 4e, Extended Data Fig. 8, Supplementary Data 3, 4). Notably, the erect bilaminate body-plan of *P. gatehousei* provides the earliest example of a colony form that has been repeatedly modified with adaptive branching structure in younger Palaeozoic bryozoans[2,7,19,22,23,27] (Extended Data Fig. 7). Although the last common ancestor of total-group Bryozoa remains enigmatic, the organic nature and basal phylogenetic position of *Protomelission* support the interpretation that crown-group Bryozoa most probably evolved from a colonial (rather than solitary) ancestor[23–25] with skeletal biomineralization independently evolving at least twice across two major bryozoan clades in post-Cambrian times; the Stenolaemata during the Early Ordovician and the Gymnolaemata (Cheilostomata) in the Jurassic period[2,5,6,19] (Fig. 4e).

The discovery of a stem bryozoan in the Cambrian narrows the origination gap that previously existed between the known fossil record and independent molecular clock estimates[11,12,21]. Our results push back the fossil record of the Bryozoa by approximately 35 million years and show that the colonial body-plan of Bryozoa can be traced back to the early Cambrian (Age 3), coincident with other major metazoan phyla belonging to the deuterostomes[14], lophotrochozoans[16,17,21] and ecdysozoans[12,25]. The miniaturized body-plan, much thinner, unmineralized cuticles (compared to arthropods and 'worms') and hard substrate habitat of early bryozoans such as *P. gatehousei* explain the poor fossil record and cryptic history of bryozoan stem taxa in the Cambrian[11,14,28]. However, the rapid diversification of the Bryozoa[6,7,30] during the Ordovician probably coincides with calcite seas[5], increasing hardground development and more robust biomineralization, leading to increased bryozoan colony size (centimetre to decimetre scale) and enhancing fossilization potential[4,5,8,11,20]. Thus, the recognized sequence of appearance for bryozoan taxa over geological time probably does not fully convey the real evolutionary history and may not provide a comprehensive understanding of bryozoan phylogeny[2,7].

The early Cambrian is recognized as an important phosphatization window for microfossil preservation[11,28] and the phosphatized stem bryozoan reported here reveals a previously hidden history for Bryozoa that provides a new framework for understanding the origin and phylogeny of the phylum[2,7]. The honeycomb-like network of zooids in *P. gatehousei* demonstrates that hierarchical architecture and complexity[24,29] of colonial life was also an important evolutionary innovation during the Cambrian radiation of animal life.

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

## Methods

### Terminology

We follow the morphological terminology used in previous studies of fossil and extant bryozoans[1,2,4,19,20].

### Material

Secondarily phosphatized specimens were recovered through standard acetic leaching of fossiliferous limestone samples, along with abundant benthic filter-feeding brachiopods[31]. Fossils were manually picked from acid residues using a binocular stereo microscope. Five incomplete specimens (Sample SADME 10470, 10470-1–10470-4) were collected from nodular, sandy limestones of the lower Wirrealpa Limestone (Cambrian Stage 4) at the Ten Mile Creek section, Bunkers Graben, South Australia[9]. One complete specimen of *P. gatehousei* (Sample ELI XYB 4 AN04) was collected from grey fossiliferous limestones in the Xihaoping Member of the Dengying Formation (Cambrian Stage 3), at the Xiaoyangba section, Hanzhong, South China. The geological and geographic setting has been previously described in detail[31].

### Scanning electron microscopy

Identified specimens were selected for the study using a Zeiss Supra 35 VP field emission at Uppsala University, Fei Quanta 450-FEGSEM at Northwest University and JEOL JSM 7100F-FESEM at Macquarie University. Coated specimens were further analysed with Backscattered electron imaging (BSE) in Quanta FEG 450 and JEOL JSM 7100F, with attached Energy Dispersive X-ray spectrometry (EDS) system, with 20.0 kV, 60 Pa and WD 11.4 mm at Northwest University and Macquarie University.

### X-ray tomographic microscopy

Two specimens were scanned using an Xradia MicroXCT-400 system (Carl Zeiss XRM) with the source operating at 80 kV, 125 µA over 180° sample rotation (−92° to 92°) at The University of Sydney. Geometric and optical magnification settings were chosen to collect projections with $xy$-pixel dimensions of 2.0334 µm (Samples SADME 10470 and ELI XYB 4 AN02). The projections were reconstructed using XMReconstructor Version 7.0.2817 (Carl Zeiss XRM) to produce a series of 16-bit TIFF images with a slice spacing equivalent to the pixel $xy$ dimensions (isotropic voxels) and voxel size of 2.03 µm. The X-ray tomographic microscopy (µCT) images were visualized and segmented via thresholding using ORS Dragonfly 324 version 2020.2 (software available at http://www.theobjects.com/dragonfly). Before feature extraction, images were applied with a normalization filter, unsharp mask and mean shift filter using the image processing function of Dragonfly. Morphological features of interest were coloured separately to assist in distinguishing them from one another. Three-dimensional videos are provided in Supplementary Videos 1–6.

### Measurements

Measurements of the length, width and angle of different parts of *P. gatehousei* were performed on µCT and SEM images by TpsDig2 v. 2.16. Scatter plots of different specimens, analysed by PAST v. 3, showing morphological variations, were also constructed. Raw data are provided in Supplementary Data 1, 2. Abbreviations used in the figures: B, basal wall; F, frontal wall; H, holdfast; M, mesotheca/median lamina; Z, zooid.

### Phylogenetic analysis

Fifty-two characters were coded for *Protomelission*, 18 bryozoan genera and 2 outgroup taxa (a total of 21 taxa). The phylogenetic data matrix was built in Microsoft Excel 2016. The 18 bryozoan genera are exemplars of the eight major bryozoan orders, and the fossil genera chosen all occur in the Ordovician (except for *Fenestrapora*, which is Devonian). The two outgroup taxa (*Eoobolus* and *Phoronis*) correspond to the two major non-bryozoan clades within the Lophophorata. Character codings were based on previously published data (Supplementary Data 3). All character codings are provided in Nexus format, along with a full list of the characters used, in Supplementary Data 3, 4.

Phylogenetic trees were inferred using both maximum parsimony and Bayesian methods. Parsimony analysis was performed using PAUP* (v. 4.0a169)[32]. A non-parametric bootstrap search based on 1,000 replicates was conducted using a heuristic search algorithm, with starting trees built using stepwise addition and branch swapping undertaken using tree bisection and reconnection (TBR). Results of this bootstrap analysis were summarized as a 50% majority rule consensus tree (Extended Data Fig. 8). Bayesian analyses were run using MrBayes (v.3.2.7)[33] and the Mkv model[34], with gamma-distributed rate variation and variable coding. The analysis used a sampling frequency of 1,000, two concurrent runs, four Metropolis-coupled chains, and was run for 10 million generations. A 25% relative burn-in was implemented for all summary statistics. The resulting phylogenetic tree is presented in Fig. 4e.

### Reporting summary

Further information on research design is available in the Nature Research Reporting Summary linked to this paper.

## Data availability

All data analysed in this study, including the phylogenetic datasets, are available in the Article, Extended Data Figs. 1–8, Extended Data Tables 1, 2 or Supplementary Information. Raw datasets are provided in the Dryad Digital Repository (https://doi.org/10.5061/dryad.rn8pk-0pbd). CT scans and parameters used for scanning of specimens in this publication can be accessed in the MorphoSource Repository (https://doi.org/10.17602/M2/M379121 and https://doi.org/10.17602/M2/M379116) and the affiliated project (https://www.morphosource.org/projects/000378949?locale=en).

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

**Acknowledgements** This study was supported by a Macquarie University Research Fellowship (2019 MQRF), National Natural Science Foundation of China (41720104002, 41621003, 41890844 and 41772010), Strategic Priority Research Program of Chinese Academy of Sciences (XDB26000000), Ministry of Science and 111 project of Ministry of Education of China (D17013), Thousand Talents Program and 1000 Talent Shaanxi Province Fellowship. We thank Cambridge University Press for permission to illustrate *P. gatehousei* in Fig. 1a, f, reproduced from ref. [9], with permission from Cambridge University Press. The palaeogeographic map in Extended Data Fig. 1 is reproduced from ref. [35] with permission from Geological Society Publishing House and Extended Data Fig. 1b is reproduced from ref. [31] with permission from Elsevier Science & Technology Journals. We thank L. E. Holmer and M. Streng at Uppsala University, Y. L. Pang at Northwest University, and S. Lindsay and C. Shen at the Microscopy Unit of Macquarie University for assistance with SEM imaging, M. Foley at University of Sydney for technical assistance with µCT imaging, X. Liu for reconstruction, Q. C. Feng for sample preparation, and J. Alroy, C. Simpson and M. Steinthorsdottir for discussion.

**Author contributions** Zhiliang Zhang, Zhifei Zhang and G.A.B. conceived the research. G.A.B. and Zhifei Zhang led fieldwork in Australia and China, respectively. Zhiliang Zhang prepared all specimens, photographs and figures. J.M, P.D.T., L.C.S. and Zhiliang Zhang undertook phylogenetic analyses. S.M.J. performed visualization with μCT data. Zhiliang Zhang and G.A.B. wrote early drafts of the paper with input from all other authors. All authors discussed the results and approved the final manuscript.

**Competing interests** The authors declare no competing interests.

**Additional information**
**Correspondence and requests for materials** should be addressed to Zhiliang Zhang, Zhifei Zhang or Luke C. Strotz.

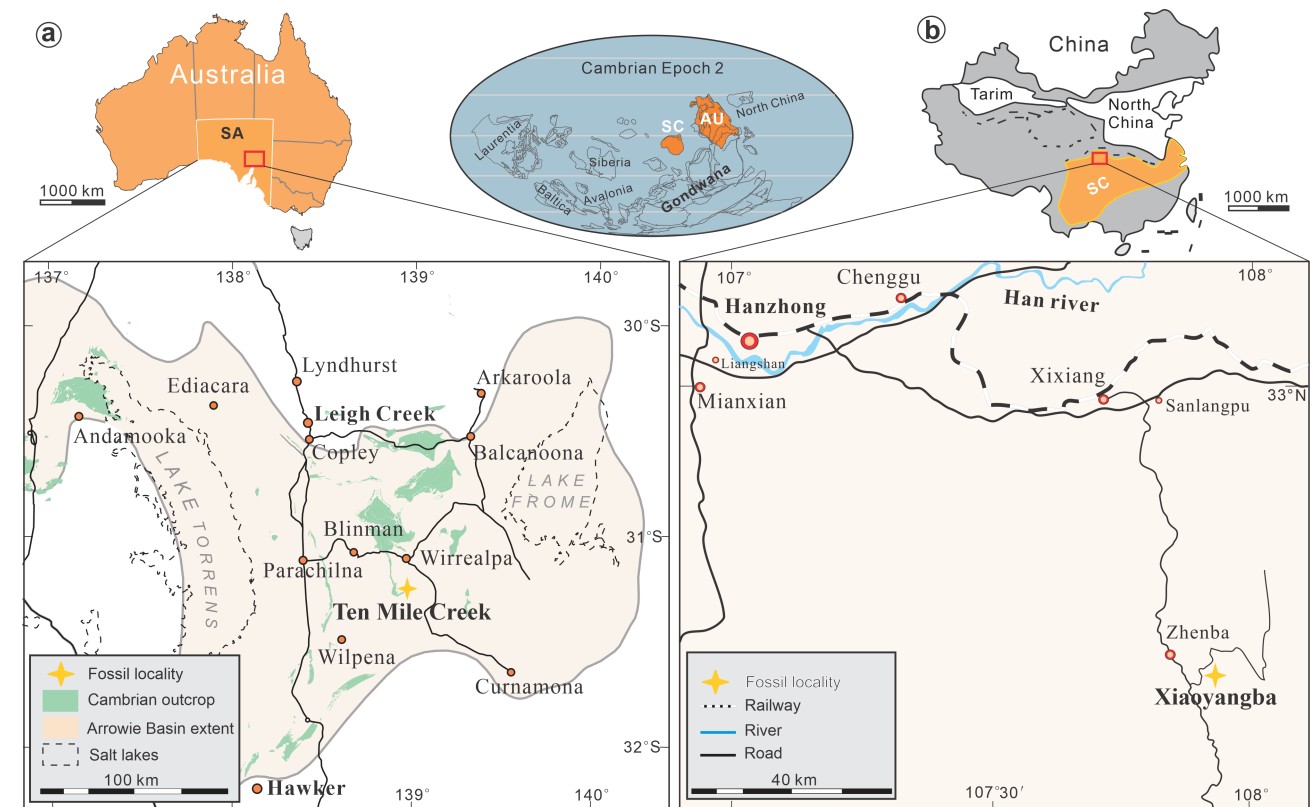

**Extended Data Fig. 1 | Geographic map of fossil localities and palaeogeographic distribution of Australia (AU) and South China (SC) platforms during the early Cambrian[35]. a**, Locality of the Ten Mile Creek section, Arrowie Basin, South Australia (SA), showing Cambrian outcrop[9,36]. **b**, Locality map of the Xiaoyangba section, Hanzhong, South China[31].

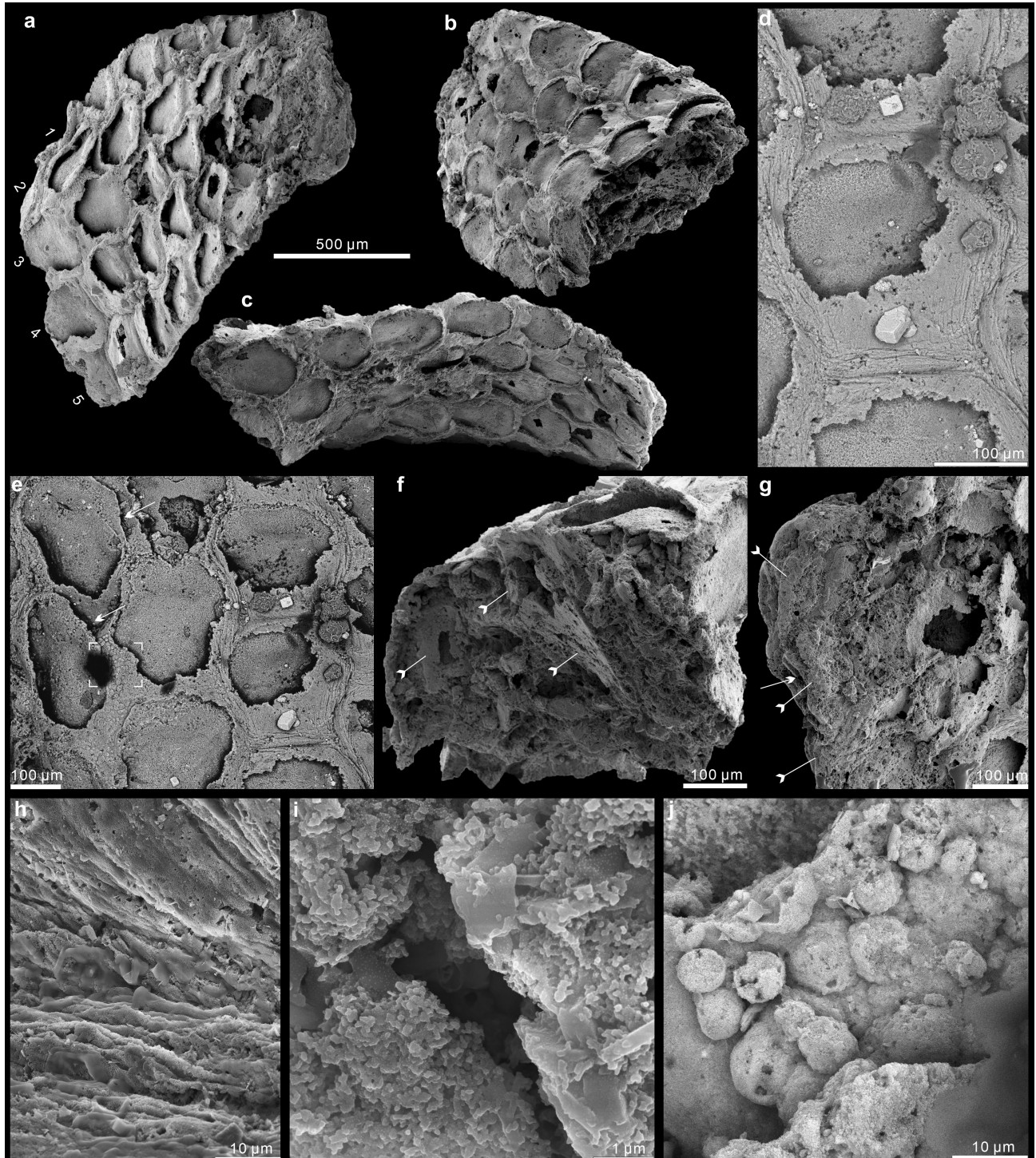

**Extended Data Fig. 2 | *Protomelission gatehousei* from the early Cambrian Xihaoping Member of Dengying Formation, South China, ELI XYB 4 AN02. a**, Oblique lateral view of the colony; note five series of zooids. **b**, Oblique basal view showing the elliptical holdfast. **c**, Oblique view of the opposite colony surface. **d**, BSE imaging demonstrates distinct wrinkles along the zooid edges. **e**, BSE imaging, arrows demonstrate the space between adjacent zooids, box corners indicate the spherulitic microstructures shown in figure j. **f**, Close-up of holdfast with tailed arrows showing attached small grains. **g**, Enlargement of weakly phosphatized colony apex; note basal walls of three adjacent zooids indicated by tailed arrows and median mesotheca by an arrow. **h**, Detail of wrinkles of frontal wall. **i**, Enlarged diagenetic apatite of figure h. **j**, Enlargement of spherulitic microstructures between adjacent zooids.

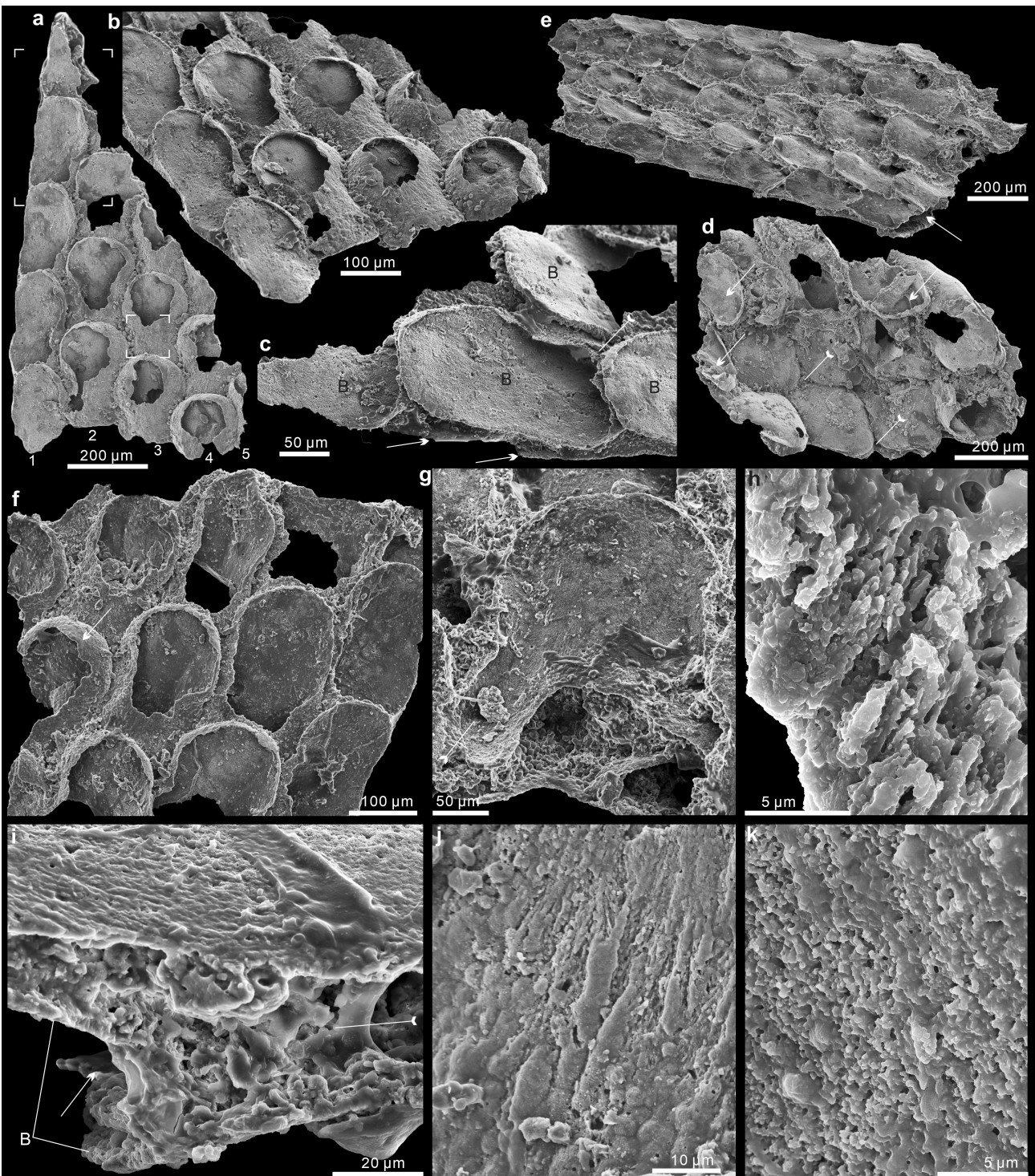

**Extended Data Fig. 3 | *Protomelission gatehousei* from the early Cambrian Wirrealpa Limestone, South Australia. a–c** and **j-k**, SADME 10470-2. **a**, Front side of a broken colony, highlighting five series of zooids, upper box corners indicate the area shown in figure c, lower box corners show the microstructures in figure j. **b**, Oblique lateral view, with relatively well preserved frontal walls and circular orifice of zooids. **c**, Lateral view showing the staggered pattern of zooids in both layers of bifoliate colony; note basal walls and largely broken frontal wall (tailed arrow), and basal walls from opposite layer (arrows). **d**, SADME 10470-3, back view of one colonial layer with the exfoliated opposite layer, showing three broken zooids from the opposite layer (arrows) and space between adjacent zooids (tailed arrows). **e-i**, Holotype, SADME 10470. **e**, Oblique lateral view with arrows showing the median mesotheca between two layers. **f**, Quincuncial arrangement of box-shaped zooids, showing dome shaped frontal wall (arrow). **g**, Detail of spoon-shaped structure at the zooid proximal end, indicated by a tailed arrow. **h**, Recrystallized granules and fibres of frontal wall. **i**, Lateral view showing one frontal wall (arrow) and two basal walls of the same layers, and the phosphatized median mesotheca (tailed arrow). **j**, Spherulitic microstructures of frontal wall. **k**, Enlarged diagenetic apatite of basal wall. B, basal wall.

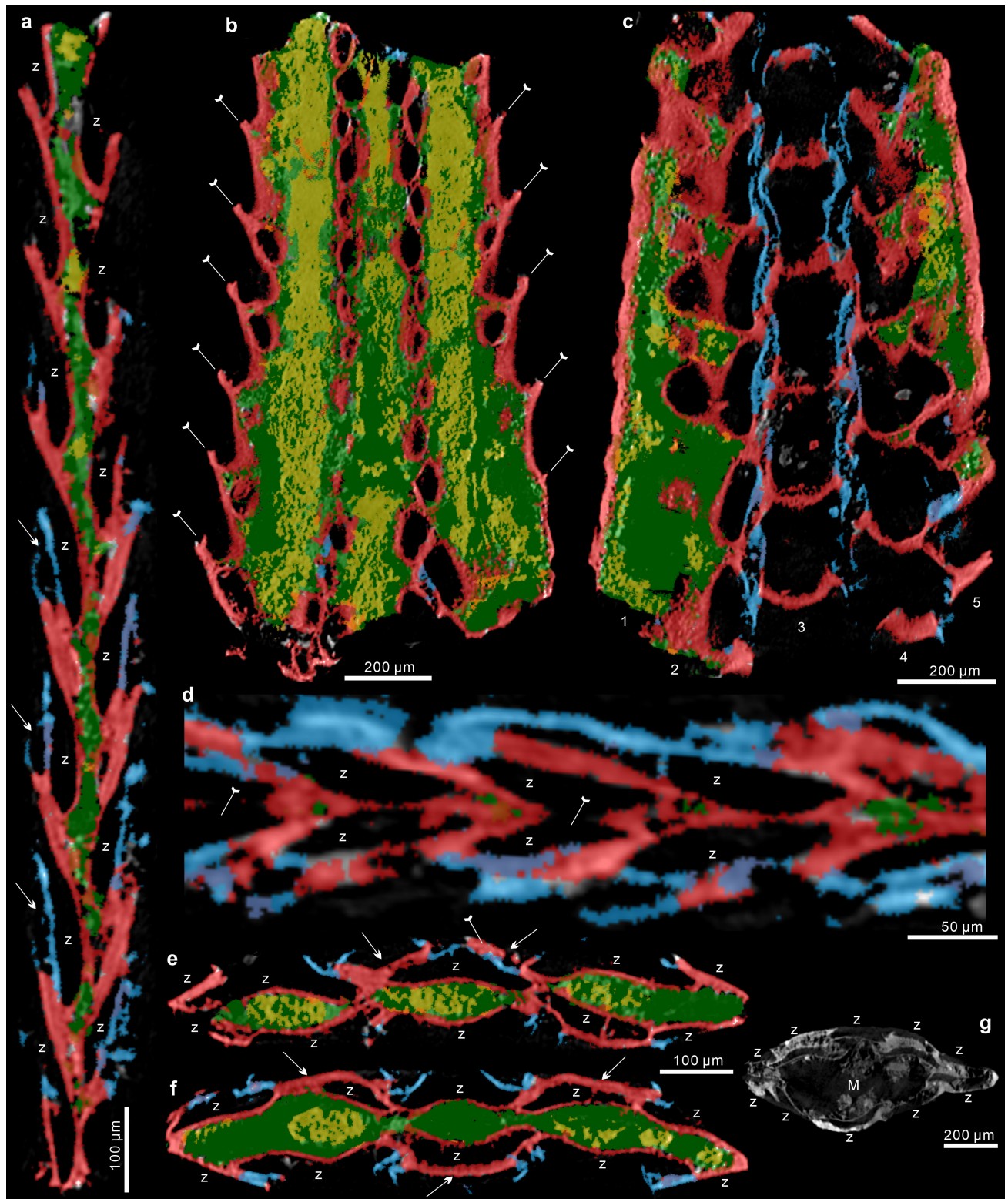

**Extended Data Fig. 4 | μCT images of *Protomelission gatehousei* from the early Cambrian of South Australia and South China. a-f**, SADME 10470. **a**, Longitudinal section, showing bifoliate pattern of zooids on the back-to-back layers; note the space between the frontal wall of parent and daughter zooids by arrows. **b-c**, Tangential section. **b**, Median mesotheca and curved basal walls by tailed arrows. **c**, Frontal walls of Series-3, demonstrating the concave centre of the back/opposite layer. **d**, Longitudinal section shows detail of probable colonial connection between adjacent zooids from back-to-back layers (tailed arrows) through space of median mesotheca. **e–f**, Transverse section showing zooids on bilaminate layers along mesotheca. **e**, New budding zooids indicated by arrows, and daughter basal wall overlapping parent frontal wall indicated by a tailed arrow. **f**, Basal walls of new budding zooids indicated by arrows. **g**, ELI XYB 4 AN02, transverse section showing zooids on both layers and median mesotheca of the colony. Blue, frontal wall; Green, mesotheca with secondary phosphatic cement in yellow; Red, basal wall. M, mesotheca; Z, zooid.

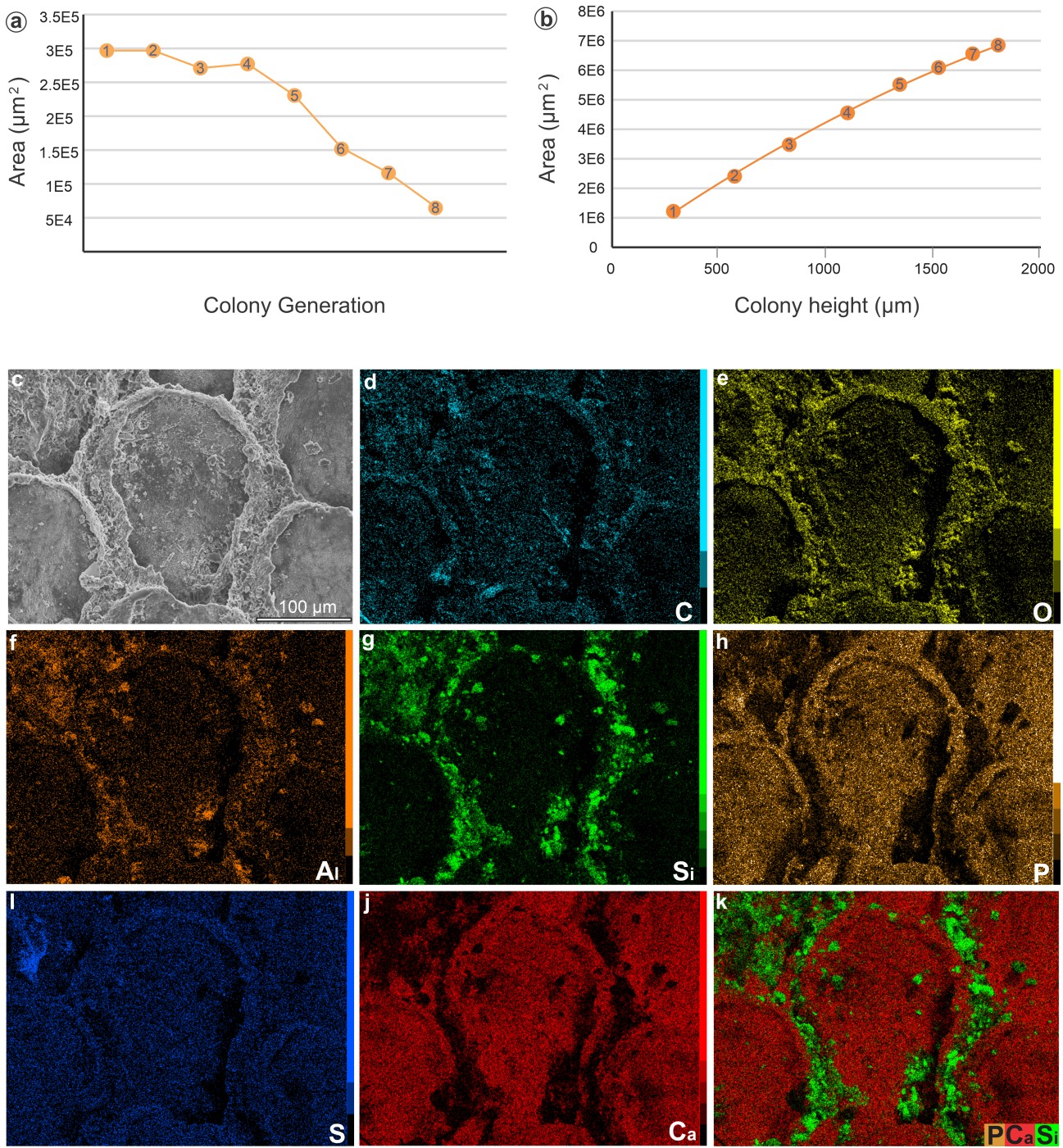

**Extended Data Fig. 5 | Relationship between colony growth and colony surface area, and elemental mapping of the zooids of *Protomelission gatehousei*. a**, Plots of increasing colony surface area and colony generations (1–8). **b**, Plots of total colony surface area and colony height, indicating a uniform increase of the colonial surface area during development. **c**–**k**, EDS elemental mapping, SADME 10470. **c**, SEM image. **d**, Elemental map of C. **e**, Elemental map of O. **f**, Elemental map of Al. **g**, Elemental map of Si. **h**, Elemental map of P. **i**, Elemental map of S. **j**, Elemental map of Ca. **k**, Elemental map of P, Ca and Si concentrations, noting the clastic particles adhered to the space between adjacent zooids.

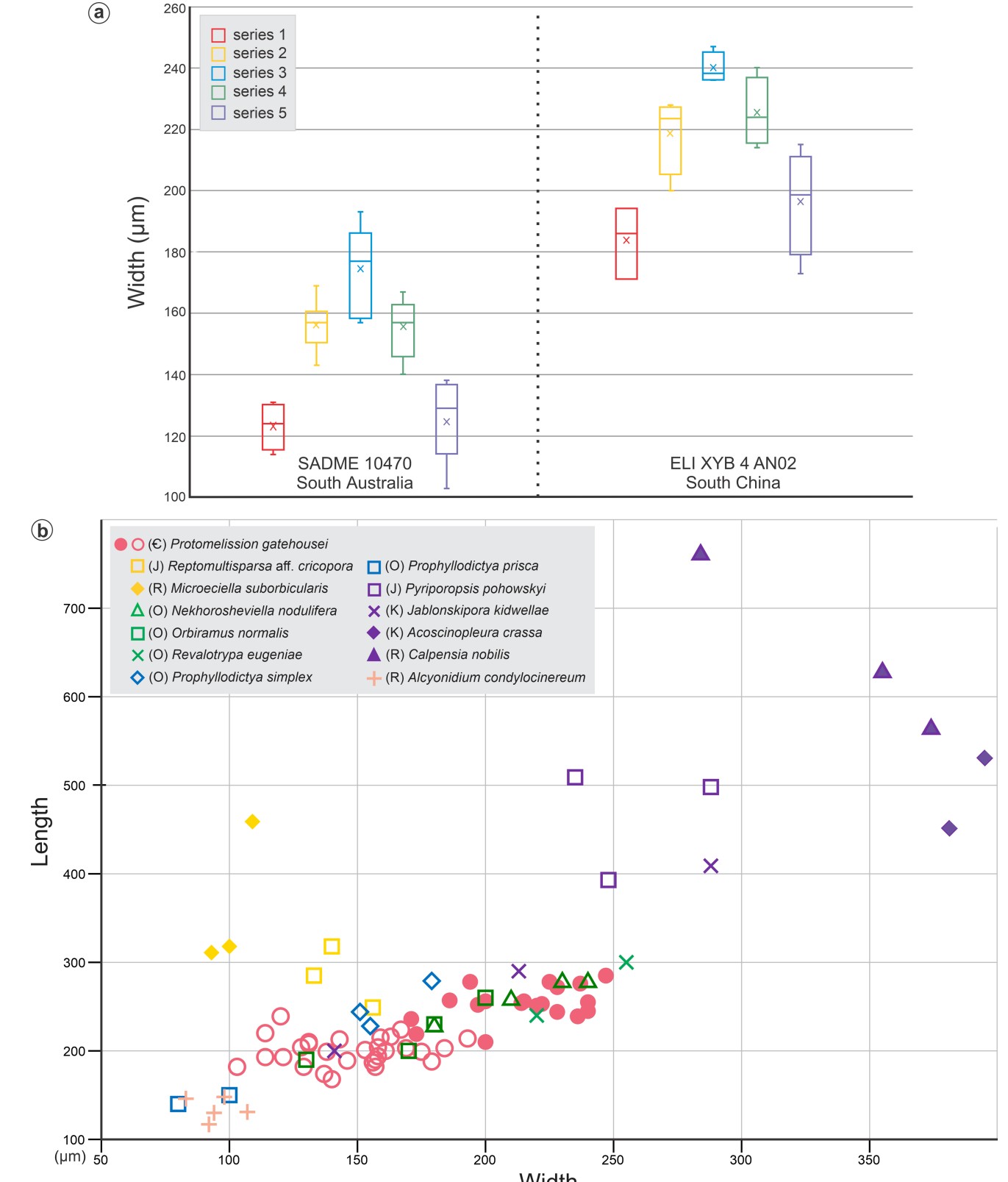

**Extended Data Fig. 6** | See next page for caption.

**Extended Data Fig. 6 | Plots of zooid size of *Protomelission gatehousei*.** Each pair of width (greatest width perpendicular to the proximal-distal axis of the zooid) and length (proximal-distal axis of the zooid) values correspond to a single zooid. **a**, Boxplots of zooid width for five adjacent series (different colours) from two *P. gatehousei* colonies from South Australia and South China, with the mean value for each series indicated by the ×. The colour of each series matches the colours used in Fig. 4d. Values for boxplots are provided in Supplementary Data 1. N = 49 biologically independent measurements of zooid size. **b**, Zooid length and width of different bryozoan taxa (different colours) from the literature with the geological age of each taxon indicated in the key. Raw data for scatter plots are provided in Supplementary Data 2. N = 172 biologically independent measurements of zooid size (86 zooids). Note that for Ordovician taxa, the size range is comparable to *Protomelission*. Red = *Protomelission gatehousei* (circle, SADME 10470; solid circle, ELI XYB 4 AN02); Yellow = Cyclostomata; Green = Trepostomata; Blue = Cryptostomata; Purple = Cheilostomata; Pink = Ctenostomata[4,5,26,37–41]. Є, Cambrian, O: Ordovician; J, Jurassic; K, Cretaceous, R: Recent.

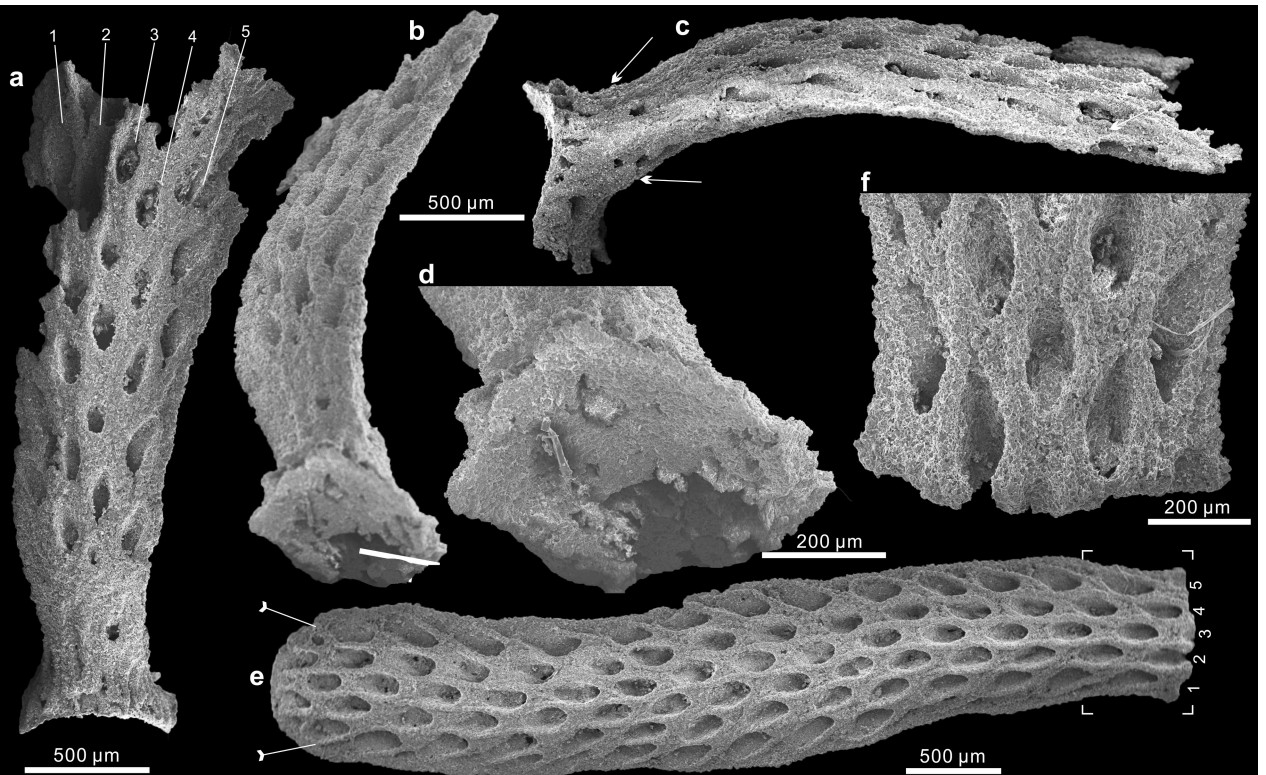

**Extended Data Fig. 7 | Erect bifoliate bryozoan *Stictopora* sp. from the Late Ordovician Bowen Park Group, New South Wales, Australia. a**–**d**, Bowen-DQ-01. **a**, Front side of the colony, noting the five series of zooids and arc-shaped holdfast. **b**, Oblique basal view. **c**, Lateral view showing zooids on both layers by arrows, and arc-shaped holdfast. **d**, Enlarged holdfast, note the relatively smooth surface as an adaptation to a hard substrate. **e**–**f**, Bowen-DQ-02. **e**, Front side of a larger colony showing five series of zooids and new budding series at the apex indicated by tailed arrows. Box corners indicate the area in figure f. **f**, Enlargement of five series of zooids.

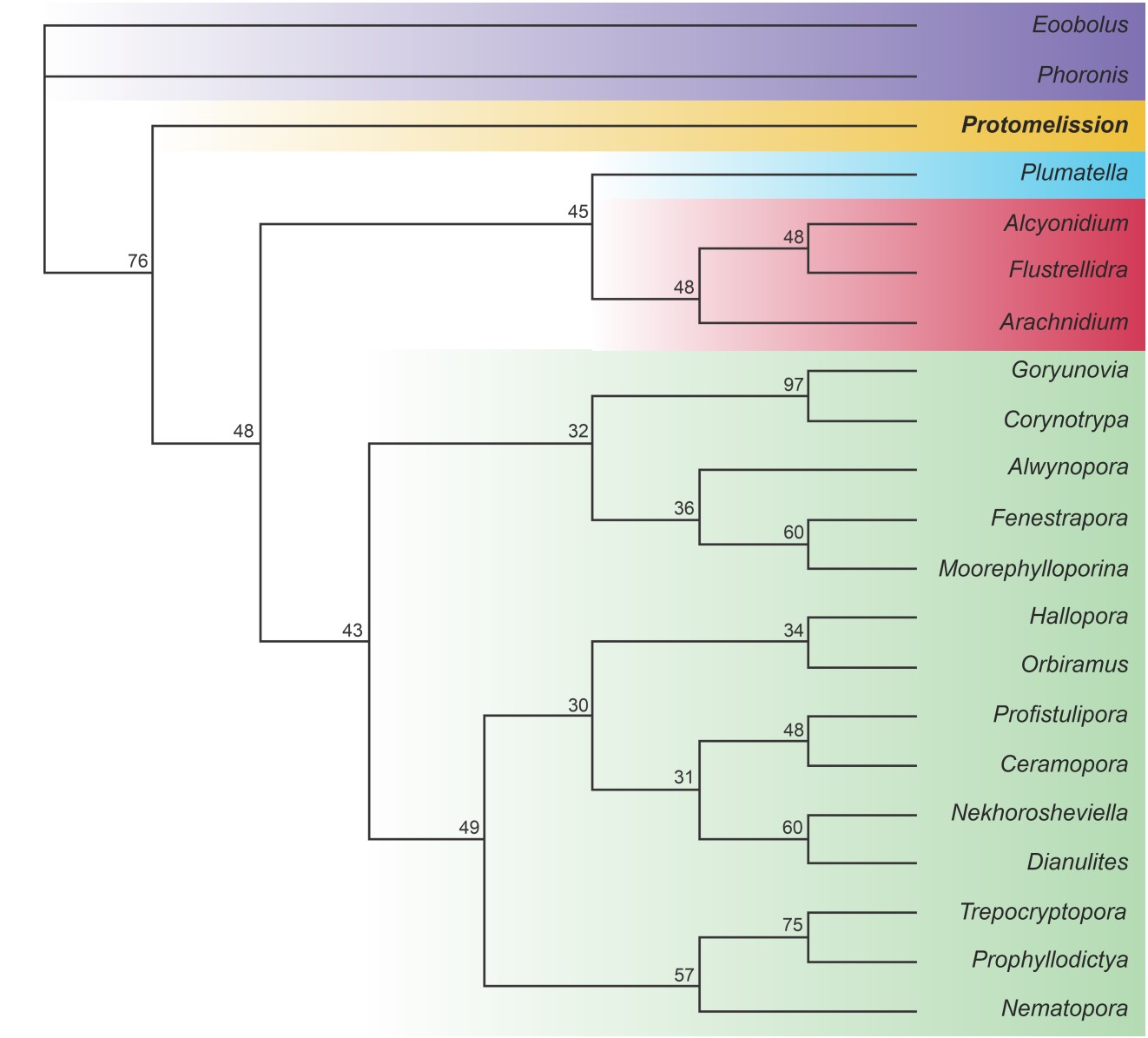

**Extended Data Fig. 8 | Phylogenetic tree inferred using parsimony based on a matrix of 21 taxa and 52 characters.** Fifty-percent majority rule bootstrap consensus tree generated using a heuristic search algorithm by PAUP*. Node values are bootstrap probabilities. Coloured areas indicate the three taxonomic classes that comprise the Bryozoa, along with the position of *P. gatehousei* and outgroups (see Supplementary Data 3-4 for details). Purple = outgroups; Yellow = *Protomelission*; Blue = Phylactolaemata; Red = Gymnolaemata; Green = Stenolaemata.

**Extended Data Table 1 | Width and length of five adjacent series of zooids of *Protomelission gatehousei***

| | | W | L | W | L | W | L | W | L | W | L |
|---|---|---|---|---|---|---|---|---|---|---|---|
| | | Series 1 | | Series 2 | | Series 3 | | Series 4 | | Series 5 | |
| SADME 10470 South Australia | | 120 | 239 | 143 | 213 | 159 | 215 | 163 | 216 | 114 | 220 |
| | | 131 | 210 | 169 | 203 | 193 | 214 | 167 | 224 | 131 | 208 |
| | | 128 | 204 | 158 | 204 | 184 | 203 | 161 | 200 | 137 | 174 |
| | | 114 | 193 | 153 | 201 | 175 | 199 | 157 | 190 | 138 | 199 |
| | | | | 158 | 194 | 179 | 188 | 156 | 187 | 129 | 182 |
| | | | | 156 | 187 | 157 | 182 | 146 | 189 | 121 | 193 |
| | | | | | | | | 140 | 168 | 103 | 182 |
| | Mean | 123 | 212 | 156 | 200 | 175 | 200 | 156 | 196 | 125 | 194 |
| | | W | L | W | L | W | L | W | L | W | L |
| ELI XYB 4 AN02 South China | | Series 1 | | Series 2 | | Series 3 | | Series 4 | | Series 5 | |
| | | 194 | 278 | 228 | 272 | 247 | 285 | 220 | 251 | 215 | 256 |
| | | 186 | 257 | 222 | 253 | 240 | 255 | 214 | 254 | 200 | 256 |
| | | 171 | 236 | 225 | 278 | 237 | 276 | 240 | 245 | 197 | 252 |
| | | | | 200 | 210 | 236 | 239 | 228 | 244 | 173 | 219 |
| | Mean | 184 | 257 | 218 | 253 | 240 | 264 | 226 | 249 | 196 | 246 |

For two colonies, one from South Australia and one from South China. Each pair of width (greatest width perpendicular to the proximal-distal axis of the zooid) and length (proximal-distal axis of the zooid) values correspond to a single zooid. N=98 biologically independent measurements of zooid size (49 zooids). The colour of each series matches the colours used in Fig. 4d. The total mean zooid width (all series combined)=174 μm and mean length=220 μm. L, length; W, width (measurements are in μm).

**Extended Data Table 2 | Character traits expected in ancestral Cambrian bryozoans[2]**

| Character traits expected in ancestral Bryozoa | *Protomelission gatehousei* | Data |
|---|---|---|
| The founding zooid of a colony, the ancestrula, is morphologically distinct from the later budded zooids, being typically smaller and less complex. | Not preserved. Rarely preserved in living erect bryozoans, due to breakage or subsequent overgrowth. | Fig. 2c; Extended Data Fig. 2f |
| Zooids of fossil bryozoans should seldom exceed 1 mm in maximum surface dimension (cf. the significantly larger modules of colonial corals). | Comfortably within the bryozoan zooid range. 174 μm in width, 220 μm in length. | Extended Data Fig. 6; Table 1 |
| The sizes, shapes and arrangement of zooids within the bryozoan colony are semi-regular; for example, colonies with geometrically perfect arrays of regular hexagonal zooids do not occur. | The pinching out of longitudinal module series disrupts perfect regularity. | Figs 1a, 2a, 3b-c |
| Each zooid has at least one main opening, aperture or orifice, to allow passage of the lophophore during life. | Clearly defined at the distal end in well preserved examples. | Fig. 1i; Extended Data Fig. 3a-b |
| Zooid opening ranges from about 50 μm to almost 1 mm in diameter. | ~50 μm in diameter. | Fig. 1i; Extended Data Fig. 3a-b |
| Depending on species, zooid shape can be box-shaped and hexagonal or rectangular in plan, or take the form of long curved tubes. | Box-shaped and hexagonal in plan. | Figs 1f, 2d, 3b; Extended Data Fig. 3e-f |
| Apart from tiny pores in the skeletal walls, zooidal skeletons are complete and fully enclose the chamber of the zooid; chambers do not bifurcate to produce new buds, unlike some colonial corals. | Zooid chambers completely bounded by walls, without bifurcation. | Figs 1h, 2f, 3d, f |
| Except for taxa lacking biomineralization, the mineralized skeletons of bryozoans are always calcareous, usually calcitic but occasionally aragonitic or bimineralic in composition. | Lacking biomineralization, secondarily phosphatized. | Fig. 2g; Extended Data Figs 2h-j, 3j-k, 5c-k |

Zhifei Zhang
Luke C. Strotz

# Reporting Summary

## Statistics

For all statistical analyses, confirm that the following items are present in the figure legend, table legend, main text, or Methods section.

| n/a | Confirmed | |
|---|---|---|
| ☐ | ☒ | The exact sample size (*n*) for each experimental group/condition, given as a discrete number and unit of measurement |
| ☐ | ☒ | A statement on whether measurements were taken from distinct samples or whether the same sample was measured repeatedly |
| ☒ | ☐ | The statistical test(s) used AND whether they are one- or two-sided *Only common tests should be described solely by name; describe more complex techniques in the Methods section.* |
| ☒ | ☐ | A description of all covariates tested |
| ☒ | ☐ | A description of any assumptions or corrections, such as tests of normality and adjustment for multiple comparisons |
| ☒ | ☐ | A full description of the statistical parameters including central tendency (e.g. means) or other basic estimates (e.g. regression coefficient) AND variation (e.g. standard deviation) or associated estimates of uncertainty (e.g. confidence intervals) |
| ☒ | ☐ | For null hypothesis testing, the test statistic (e.g. *F*, *t*, *r*) with confidence intervals, effect sizes, degrees of freedom and *P* value noted *Give P values as exact values whenever suitable.* |
| ☐ | ☒ | For Bayesian analysis, information on the choice of priors and Markov chain Monte Carlo settings |
| ☒ | ☐ | For hierarchical and complex designs, identification of the appropriate level for tests and full reporting of outcomes |
| ☒ | ☐ | Estimates of effect sizes (e.g. Cohen's *d*, Pearson's *r*), indicating how they were calculated |

*Our web collection on statistics for biologists contains articles on many of the points above.*

## Software and code

Policy information about availability of computer code

| | |
|---|---|
| Data collection | XMReconstructor v. 7.0.2817 and ORS Dragonfly, v. 2020.2 for µCT reconstruction, visualisation and segmentation. TpsDig2 v. 2.16 for size measurement. Microsoft Excel 2016 for size data and coding phylogenetic matrices. |
| Data analysis | MrBayes v.3.2.7 (open source) and PAUP* v. 4.0a169 (freely available from Phylosolutions) for phylogenetic analyses. The settings needed to replicate these analyses are provided in the paper. |

For manuscripts utilizing custom algorithms or software that are central to the research but not yet described in published literature, software must be made available to editors and reviewers. We strongly encourage code deposition in a community repository (e.g. GitHub). See the Nature Portfolio guidelines for submitting code & software for further information.

## Data

Policy information about availability of data

All manuscripts must include a data availability statement. This statement should provide the following information, where applicable:

- Accession codes, unique identifiers, or web links for publicly available datasets
- A description of any restrictions on data availability
- For clinical datasets or third party data, please ensure that the statement adheres to our policy

All data analysed in this paper, including the phylogenetic datasets, are available as part of the Article, Extended Data Figs. 1–8, Extended Data Tables 1-2, Supplementary Information. CT scans and parameters used for scanning are available in the MorphoSource Repository (https://www.morphosource.org/concern/media/000379116 and https://www.morphosource.org/concern/media/000379121). Raw datasets are available in the Dryad Digital Repository (https://doi.org/10.5061/dryad.rn8pk0pbd).

# Field-specific reporting

Please select the one below that is the best fit for your research. If you are not sure, read the appropriate sections before making your selection.

☐ Life sciences ☐ Behavioural & social sciences ☒ Ecological, evolutionary & environmental sciences

For a reference copy of the document with all sections, see nature.com/documents/nr-reporting-summary-flat.pdf

# Ecological, evolutionary & environmental sciences study design

All studies must disclose on these points even when the disclosure is negative.

| | |
|---|---|
| Study description | This is a palaeontological and taxonomic study including collection, preparation, microscopy imaging, description and phylogenetic analyses of fossil material from lower Cambrian rocks of South Australia and South China. |
| Research sample | The specimens used represent all currently known specimens of P. gatehousei (including the holotype) and illustrate all the key taxonomic features of this species. The are more than adequate to establish that P. gatehousei is a stem-group bryozoan |
| Sampling strategy | We utilise all currently known specimens of P. gatehousei (including the holotype). These specimens provide all the necessary information to establish the evolutionary affinities of P. gatehousei. |
| Data collection | Glenn Brock led the fossil excavation at the Ten Mile Creek section and found the holotype specimen (SADME 10470) in 1987, and Zhiliang Zhang recovered four paratype specimens (SADME 10470-1—10470-4) from acid macerated residues in 2019. Zhifei Zhang, Zhiliang Zhang and Feiyang Chen undertook fossil excavation at the Xiaoyangba section, and Zhiliang Zhang discovered ELI XYB 4 AN04. SEM, BSE and EDS images were collected using Zeiss Supra 35 VP field emission, Fei Quanta 450-FEGSEM and JEOL JSM 7100F-FESEM. µCT projections were collected using a Xradia MicroXCT-400 system. Measurements of the length, width and angle of different parts of P. gatehousei were performed on µCT and SEM images. |
| Timing and spatial scale | Collection of the specimens of P. gatehousei from the Ten Mile Creek section took place in 1987. Collection of the specimens from the Xiaoyangba section took place in 2015. The material collected, which represents all of the material of P. gatehousei currently known, is more than adequate to establish the evolutionary affinities of P. gatehousei. |
| Data exclusions | No data was excluded. |
| Reproducibility | This is a palaeontological study that utilises fossils that form part of the evolutionary record of this planet. They cannot be replicated or duplicated. We provide ample information in our paper for anyone to resample the localities that are the sources of these specimens and to repeat the methods of analysis we use. |
| Randomization | no randomization was used. |
| Blinding | This is a palaeontological study of all known material of a species. Blinding is inapplicable and irrelevant. |

Did the study involve field work? ☒ Yes ☐ No

## Field work, collection and transport

| | |
|---|---|
| Field conditions | Lower Cambrian rocks were well-exposed and fossiliferous limestones were excavated manually in the field. The Wirrealpa Limestone has been weathered under a semi-arid climate with an average temperature of 25°C and an average amount of rainfall of 250 mm per year.  The Xihaoping Member of the Dengying Formation is exposed in an area with a mild and humid climate with an average temperature of 14°C and an average amount of rainfall of 1323 mm per year. |
| Location | Ten Mile Creek section, Flinders Ranges, Australia:  31°15'43" S, 138°53'15" E<br>Xiaoyangba section, Hanzhong, China: 32°29'28" N, 107°7'10" E. |
| Access & import/export | All fossil specimens in this study were collected by the field group led by Macquarie University and Northwest University in compliance with all local, national and international laws. All collecting permissions were obtained before the collecting started. Any collecting in private land also obtained permissions from the land holder.<br><br> No permits were required to collect any of the samples included in our study. |
| Disturbance | All sampling was collected by hand with minimal disturbance of the surrounding environment. |

# Reporting for specific materials, systems and methods

We require information from authors about some types of materials, experimental systems and methods used in many studies. Here, indicate whether each material, system or method listed is relevant to your study. If you are not sure if a list item applies to your research, read the appropriate section before selecting a response.

## Materials & experimental systems

| n/a | Involved in the study |
|-----|----------------------|
| ☒ | Antibodies |
| ☒ | Eukaryotic cell lines |
| ☒ | Palaeontology and archaeology |
| ☒ | Animals and other organisms |
| ☒ | Human research participants |
| ☒ | Clinical data |
| ☒ | Dual use research of concern |

## Methods

| n/a | Involved in the study |
|-----|----------------------|
| ☒ | ChIP-seq |
| ☒ | Flow cytometry |
| ☒ | MRI-based neuroimaging |

## Palaeontology and Archaeology

Specimen provenance
> Specimens were collected from the lower Wirrealpa Limestone at the Ten Mile Creek section, Bunkers Graben of Flinders Ranges, Australia, and from the Xihaoping Member of the Dengying Formation, at the Xiaoyangba section of Hanzhong City, China.
>
> No permits were required to collect these specimens

Specimen deposition
> SADME 10470 and SADME 10470—110470-4 are deposited at the South Australian Geological Survey. ELI XYB 4 AN04 is deposited at Northwest University.

Dating methods
> no new dates are provided in the paper.

☐ Tick this box to confirm that the raw and calibrated dates are available in the paper or in Supplementary Information.

Ethics oversight
> No ethics permissions were required to undertake our study.

Note that full information on the approval of the study protocol must also be provided in the manuscript.

