## [Peer Review File · Nature]

Manuscript Title:

Reviewer Comments & Author Rebuttals

Reviewer Reports on the Initial Version:

Referee #1 (Remarks to the Author):

Review of "Fossil evidence unveils an early Cambrian origin for Bryozoa" (Manuscript 2021-08-12483)

This is one of the most exciting manuscripts I have ever reviewed. I quickly agreed to be a reviewer because I just couldn't wait to read it -- and the paper did not disappoint. It is thorough, convincing, and well written. The bryozoology world has been waiting for specimens like *Protomelissia gatehousei* to illuminate stem-group bryozoans in the Cambrian, and paleontologists in general will be thrilled to see bryozoans legitimately and persuasively placed in the early Cambrian. For that matter, molecular biologists interested in molecular clock calculations of phylum origins will love this paper. The data presented, including the cool 3-D videos, overwhelmingly supports the hypothesis that *Protomelissia gatehousei* is a stem-group bryozoan. Nature is the most appropriate journal for this manuscript. I encourage publication as quickly as can be arranged.

My comments below, by line number, are minor.

52: Correct spelling is *Pywackia*. Why not include Taylor et al. (2013) in the references? It still is the most detailed analysis of why *Pywackia* is not a bryozoan. Taylor, P. D., Berning, B., and Wilson, M. A. (2013). Reinterpretation of the Cambrian "bryozoan" *Pywackia* as an octocoral. *J. Paleontol.* 87, 984–990. doi: 10.1666/13-029. [Disclosure: I am a co-author.]

187: I would like "calcite enriched oceans" expressed instead as "Calcite Seas", which then connects readers to a diverse set of literature on marine carbonate precipitation systems over geological time. This is a chance to educate scientists on the biotic and abiotic controls over biomineralization with a reference or two to the Calcite Seas literature. I would recommend one or both of these additional references -- Balthasar, U. and Cusack, M., 2015. Aragonite-calcite seas — Quantifying the gray area. *Geology*, 43(2), pp.99-102. Porter SM (2010) Calcite and aragonite seas and the de novo acquisition of carbonate skeletons. *Geobiology* 8(4):256–277.

190: "Thus, the recognised sequence of appearance for bryozoan taxa over geological time likely does not reflect the real evolutionary history and may not provide a satisfactory understanding of bryozoan phylogeny." This seems a bit harsh to me. The appearance of taxa over geological time certainly must "reflect", however dimly, some aspects of evolutionary history. Maybe instead of "does not reflect" the authors would consider "does not fully convey" or something similar? I want to feel I'm still contributing to evolutionary history by sorting out fossils over time! I agree that taxa-over-time will never "provide a satisfactory understanding of bryozoan phylogeny", but it contributes to the analysis.

The "Supplementary Video/Audio - Reconstructed 3D object of SADME 10470" did not work for me, but the other videos were fantastic.

Again, I highly recommend publication of this excellent manuscript.

Mark A. Wilson
The College of Wooster

Referee #2 (Remarks to the Author):

This is an outstanding contribution that will be regarded as a milestone in bryozoan research. The finding of a fossil bryozoan in the Cambrian period will have a high impact both in the paleontological and zoological literatures.

The summary is complete and provides the key results of this study in 16 lines. The data are sound and the methodology is correct; the conclusions are solid. The text is clear and the illustrations are of high quality. This kind of data do not require much statistics, just tables of measurements and box plots. The parsimony and Bayesian analyses were performed with an adequate number of coded characters, replicates and generations. The reference list is complete and updated, giving appropriate credit to previous work.

I don't have major criticisms to this study. In my opinion, this manuscript should be published almost as it is, after the correction of some minor mistakes:

- (1) "Spherulitic" in the text (line 120), but "spharulitic" on lines 11 and 14 of Extended Data Figure 2, and on line 29 of Extended Data Figure 3.
- (2) I don't see the solid triangles of *Calpensia nobilis* in Extended Data Figure 6.
- (3) bifoiate  bifoliate, in "List of characters and state descriptions for phylogenetic analysis".
- (4) lineal  linear, in "List of characters and state descriptions for phylogenetic analysis".
- (5) exzone  exozone, in "List of characters and state descriptions for phylogenetic analysis".
- (6) Fateral  Lateral, in "List of characters and state descriptions for phylogenetic analysis".
- (7) SADEM  SADME, on lines 18 and 22 of Extended Data Figure 3.

Juan López-Gappa

Referee #3 (Remarks to the Author):

Dear Authors,

your manuscript contains very interesting information about the origin of the phylum Bryozoa in the early Cambrian. The presented description and interpretation of the fossil *Protomelission gatehousei* Brock and Cooper, 1993 deliver an important insight into the early radiation during the Cambrian and will certainly deserve attention of the scientific community.

This manuscript deals with description and interpretation of a fossil from the early Cambrian of Australia and China interpreted being a potential stem-group bryozoan. Bryozoans are one of the most successful (but often neglected!) groups of benthic animals, with ca. 15,000 extant species flourishing in various aquatic biotopes, both fresh water and marine. The question of the origin of the phylum Bryozoa is enigmatic, because of some reasons such as small size of probable organisms as well as most probably lack of their skeletal mineralization which would prevent fossilisation. In the past several candidates for probable bryozoan ancestors in the Cambrian were presented, but they all were proved to belong to other phyla or to be even pseudofossils. Specialists were searching for the prove of the Cambrian origin of Bryozoa for a long time.

The presented description as well as the skilful reconstruction of the fossil *Protomelission gatehousei* Brock and Cooper, 1993 seem to show the most reliable candidate for an early Cambrian bryozoan. It is also supported by the presented phylogenetic analysis. This report is interesting for broad scientific community because it provides a new insight in the evolution of the Metazoa at the beginning of the diversification in the Cambrian epoch. Statistical, graphical and technical methods used in this study are absolutely appropriate and on the current stage of modern science.

However, I am a bit sceptical about the statement that bryozoans must have taken their origin from a colonial ancestor (lines 174-176). *Protomelission gatehousei* displays already complex colony form, but it does not necessary mean that all the early bryozoans were colonial. There is still a time gap between the known fossil record and calculated origin of Bryozoa (44 Ma, line 42) and appearance of *Protomelission gatehousei* (35 Ma, line 181)! According to the description and reconstructions, *Protomelission gatehousei* is not the simplest bryozoan, I would expect simpler forms before it, which could not be necessary all colonial.

The manuscript does not reveal any recognizable flaws. Furthermore, on my knowledge, no interest conflicts exist in regard to the content of the manuscript. Some few comments and suggested changes are inserted in the attached PDF files

Manuscript text:

lines 78, 158: "spoon-shape" – I would prefer "spoon-shaped"

line 123. delete "within"

lines 133-134: I would suggest insertion of a citation describing the function of palmate colonies, e.g., McKinney and Jackson, 1989.

line 218: demonstrates that ..

In the reference list, taxon names should be in Italics: lines 233, 242.

In some titles, lower case should be used: lines 246-247, 269, 366, 371,

line 284: delete "Lidgard-1985-"

line 302: the lower *Wirrealpa* Limestone ..

Extended Data Figure 3, line 29: Spherulitic, not Spharulitic!

Extended Data Figure 7: *Stictopora*, not *Strictopora*!

List of characters and state descriptions for phylogenetic analysis

Character 8: change Bioliolate to Bifoliate!

Character 15: change Linar to Linear!

Character 18: change exzone to exozone!

Character 50: change Lineal to Linear!

Character 51: change Fateral to Lateral!

References:

line 29 *Fenestrapora* (talics)

line 31: delete "Lidgard-1985-"

Sincerely yours,
Dr. Andrej Ernst

Author Rebuttals to Initial Comments:

Referee #1 (Remarks to the Author):

Review of "Fossil evidence unveils an early Cambrian origin for Bryozoa" (Manuscript 2021-08-12483)s

This is one of the most exciting manuscripts I have ever reviewed. I quickly agreed to be a reviewer because I just couldn't wait to read it -- and the paper did not disappoint. It is thorough, convincing, and well written. The bryozoology world has been waiting for specimens like *Protomelission gatehousei* to illuminate stem-group bryozoans in the Cambrian, and palaeontologists, in general, will be thrilled to see bryozoans legitimately and persuasively placed in the early Cambrian. For that matter, molecular biologists interested in molecular clock calculations of phylum origins will love this paper. The data presented, including the cool 3-D videos, overwhelmingly supports the hypothesis that *Protomelission gatehousei* is a stem-group bryozoan. Nature is the most appropriate journal for this manuscript. I encourage publication as quickly as can be arranged.

Response: We thank Referee #1 for his great interest in our research and strong recommendation for publication.

My comments below, by line number, are minor.

52: Correct spelling is *Pywackia*. Why not include Taylor et al. (2013) in the references? It still is the most detailed analysis of why *Pywackia* is not a bryozoan. Taylor, P. D., Berning, B., and Wilson, M. A. (2013). Reinterpretation of the Cambrian "bryozoan" *Pywackia* as an octocoral. *J. Paleontol.* 87, 984–990. doi: 10.1666/13-029. [Disclosure: I am a co-author.]

Response: Thanks to Ref #1 for picking up the misspelling of *Pywackia*, this has been corrected. We agree that Taylor et al. 2013 is a very important paper discussing the affinity of *Pywackia*, but given the limit of 30 references, the more recent papers (Taylor and Waeschenbach, 2015; Ma et al. 2015; Hageman and Ernst, 2019) that adequately address this debate, and that these more recent papers also refer back to the older Taylor et al. (2013) reference, we do not feel it needs to be included in our citation list. The references used are also cited more than once in the main manuscript, so cannot be "swapped out". Since the bryozoan affinity of *Pywackia* is not the focus in our current study and there is a lack of space, we have chosen not to cite this paper. (line 52)

187: I would like "calcite enriched oceans" expressed instead as "Calcite Seas", which then connects readers to a diverse set of literature on marine carbonate precipitation systems over geological time. This is a chance to educate scientists on the biotic and abiotic controls over biomineralization with a reference or two to the Calcite Seas literature. I would recommend one or both of these additional references -- Balthasar, U. and Cusack, M., 2015. Aragonite-calcite seas — Quantifying the gray area. *Geology*, 43(2), pp.99-102. Porter SM (2010) Calcite and aragonite seas and the de novo acquisition of carbonate skeletons. *Geobiology* 8(4):256–277.

Response: We agree and have changed this to "Calcite Seas" in the text. However, as discussed above, there is a lack of space to include a "once only" citation for either of the recommended papers.

The Taylor et al. (2015) reference [5] is now cited more specifically since it discusses the concept of Calcite Seas and refers directly to Porter et al. 2010 as a primary source (p. 1138). Whilst we would like to include the additional recommended citations, they are not directly related to the main theme of the text and because of the lack of space, we have not included these references. (lines 134)

190: “Thus, the recognised sequence of appearance for bryozoan taxa over geological time likely does not reflect the real evolutionary history and may not provide a satisfactory understanding of bryozoan phylogeny.” This seems a bit harsh to me. The appearance of taxa over geological time certainly must "reflect", however dimly, some aspects of evolutionary history. Maybe instead of "does not reflect" the authors would consider "does not fully convey" or something similar? I want to feel I'm still contributing to evolutionary history by sorting out fossils over time! I agree that taxa-over-time will never "provide a satisfactory understanding of bryozoan phylogeny", but it contributes to the analysis.

Response: We have softened the text to “the recognised sequence of appearance for bryozoan taxa over geological time likely does not fully convey the real evolutionary history and may not provide a comprehensive understanding of bryozoan phylogeny”. (lines 137-139)

The "Supplementary Video/Audio - Reconstructed 3D object of SADME 10470" did not work for me, but the other videos were fantastic.

Response: Sorry for the confusion. The new 3D video file is provided in the Supplementary Information. (SI lines 55-57)

Again, I highly recommend publication of this excellent manuscript.

**Mark A. Wilson
The College of Wooster**

Referee #2 (Remarks to the Author):

This is an outstanding contribution that will be regarded as a milestone in bryozoan research. The finding of a fossil bryozoan in the Cambrian period will have a high impact both in the paleontological and zoological literatures.

The summary is complete and provides the key results of this study in 16 lines. The data are sound and the methodology is correct; the conclusions are solid. The text is clear and the illustrations are of high quality. This kind of data does not require much statistics, just tables of measurements and box plots. The parsimony and Bayesian analyses were performed with an adequate number of coded characters, replicates and generations. The reference list is complete and updated, giving appropriate credit to previous work.

Response: We thank Referee #2 for his very positive comments on our paper and recommendation for publication.

I don't have major criticisms to this study. In my opinion, this manuscript should be published almost as it is, after the correction of some minor mistakes:

(1) "Spherulitic" in the text (line 120), but "spharulitic" on lines 11 and 14 of Extended Data Figure 2, and on line 29 of Extended Data Figure 3.

Response: Corrected to “spherulitic” throughout the entire manuscript. (lines 401, 404, 419)

(2) I don't see the solid triangles of *Calpensia nobilis* in Extended Data Figure 6.

Response: Many thanks for picking up this oversight. We have corrected and revised the Extended Data Fig. 6b using solid triangles, according to the figure caption. (Extended Data Fig. 6b)

(3) bifoiate  bifoliate, in "List of characters and state descriptions for phylogenetic analysis".

Response: Corrected and revised as suggested. (SI lines 401, 404, 419)

(4) lineal  linear, in "List of characters and state descriptions for phylogenetic analysis".

Response: Corrected and revised as suggested. (SI Data 9, Character 8)

(5) exzone  exozone, in "List of characters and state descriptions for phylogenetic analysis".

Response: Corrected and revised as suggested. (SI Data 9, Character 18)

(6) Fateral  Lateral, in "List of characters and state descriptions for phylogenetic analysis".

Response: Correct and changed to Frontal. (SI Data 9, Character 51)

(7) SADEM  SADME, on lines 18 and 22 of Extended Data Figure 3.

Response: Corrected and revised as suggested. (lines 408, 412)

Juan López-Gappa

Referee #3 (Remarks to the Author):

Dear Authors,

your manuscript contains very interesting information about the origin of the phylum Bryozoa in the early Cambrian. The presented description and interpretation of the fossil *Protomelission gatehousei* Brock and Cooper, 1993 deliver an important insight into the early radiation during the Cambrian and will certainly deserve attention of the scientific community.

This manuscript deals with description and interpretation of a fossil from the early Cambrian of Australia and China interpreted being a potential stem-group bryozoan. Bryozoans are one of the most successful (but often neglected!) groups of benthic animals, with ca. 15,000 extant species flourishing in various aquatic biotopes, both fresh water and marine. The question of the origin of the phylum Bryozoa is enigmatic, because of some reasons such as small size of probable organisms as well as most probably lack of their skeletal mineralization which would prevent fossilisation. In the past several candidates for probable bryozoan ancestors in the Cambrian were presented, but they all were proved to belong to other phyla or to be even pseudofossils. Specialists were searching for the prove of the Cambrian origin of Bryozoa for a long time.

The presented description as well as the skilful reconstruction of the fossil *Protomelission gatehousei* Brock and Cooper, 1993 seem to show the most reliable candidate for an early Cambrian bryozoan. It is also supported by the presented phylogenetic analysis. This report is interesting for broad scientific community because it provides a new insight in the evolution of the Metazoa at the beginning of the diversification in the Cambrian epoch. Statistical, graphical and technical methods used in this study are absolutely appropriate and on the current stage of modern science.

Response: We thank Referee #3 for his positive comments and response to our interpretation of *P. gatehousei* as a potential stem-group bryozoan from lower Cambrian in Australia and China.

However, I am a bit sceptical about the statement that bryozoans must have taken their origin from a colonial ancestor (lines 174-176). *Protomelission gatehousei* displays already complex colony form, but it does not necessarily mean that all the early bryozoans were colonial. There is still a time gap between the known fossil record and calculated origin of Bryozoa (44 Ma, line 42) and appearance of *Protomelission gatehousei* (35 Ma, line 181)! According to the description and reconstructions, *Protomelission gatehousei* is not the simplest bryozoan, I would expect simpler forms before it, which could not be necessary all colonial.

Response: We agree that there may well be an even simpler (possibly even unimodal) body plan prior to *P. gatehousei*, but if the last common ancestor of Total-group Bryozoa was a solitary form, the fossil record is mute on such an occurrence and so this must be treated as speculation. Our current study demonstrates that *P. gatehousei* has a mix of character traits that accords with the view that crown-group members of the phylum were derived from a colonial ancestor in the early Cambrian.

In light of the comments of Referee #3, we have changed the sentence in question to read: “Whilst the last common ancestor of total-group Bryozoa remains enigmatic, the organic nature and basal phylogenetic position of *Protomelission* supports the interpretation that crown-group Bryozoa most likely evolved from a colonial (rather than solitary) ancestor”. (lines 120-122)

The manuscript does not reveal any recognizable flaws. Furthermore, on my knowledge, no interest conflicts exist in regard to the content of the manuscript. Some few comments and suggested changes are inserted in the attached PDF files.

Response: We appreciate the very careful checking of the manuscript and suggested revisions by Referee #3. All are accepted and revised accordingly.

Manuscript text:

lines 78, 158: "spoon-shape" – I would prefer "spoon-shaped"

Response: Corrected and revised throughout the entire manuscript. (lines 226, 252)

line 123. delete "within"!

Response: Corrected and revised as suggested. (line 92)

lines 133-134: I would suggest insertion of a citation describing the function of palmate colonies, e.g., McKinney and Jackson, 1989.

Response: Agree. The citation has been added. (line 103)

line 218: demonstrates that.

Response: Corrected and revised as suggested. (line 143)

In the reference list, taxon names should be in Italics: lines 233, 242.

Response: Corrected and revised as suggested. (lines 159, 168)

In some titles, lower case should be used: lines 246-247, 269, 366, 371,

Response: Corrected and revised as suggested. (lines 172-173, 195, 350, 355-356)

line 284: delete "Lidgard-1985-"

Response: Corrected and revised as suggested. (line 210)

line 302: the lower Wirrealpa Limestone.

Response: Corrected and revised as suggested. (line 276)

Extended Data Figure 3, line 29: Spherulitic, not Spharulitic!

Response: Corrected and revised as suggested. (line 419)

Extended Data Figure 7: *Stictopora*, not *Strictopora*!

Response: Corrected and revised as suggested. (line 454)

List of characters and state descriptions for phylogenetic analysis

Character 8: change Bioliolate to Bifoliate!

Character 15: change Linar to Linear!
Character 18: change exzone to exozone!
Character 50: change Lineal to Linear!
Character 51: change Fateral to Lateral!

Response: Corrected and revised as suggested. Character 51 was changed to Frontal+Distal. (SI Data 9, Character 8, Character 15, Character 18, Character 50, Character 51)

References:

line 29 Fenestrapora (talics)

line 31: delete "Lidgard-1985-"

Response: Corrected and revised as suggested. (SI lines 100,102)

Sincerely yours,
Dr. Andrej Ernst